# The Removal of Erythromycin and Its Effects on Anaerobic Fermentation

**DOI:** 10.3390/ijerph19127256

**Published:** 2022-06-14

**Authors:** Huayong Zhang, Meixiao Yin, Shusen Li, Shijia Zhang, Guixuan Han

**Affiliations:** Research Center for Engineering Ecology and Nonlinear Science, North China Electric Power University, Beijing 102206, China; yin_meixiao@163.com (M.Y.); lishusen@shwpg.com (S.L.); zhangsj@ncepu.edu.cn (S.Z.); 120192232409@ncepu.edu.cn (G.H.)

**Keywords:** antibiotics, livestock manure, anaerobic fermentation, substrate biodegradation, methanogens

## Abstract

In view of the problems of antibiotic pollution, anaerobic fermentation technology was adopted to remove erythromycin in this study. The removal of erythromycin and its effects mechanism on anaerobic fermentation were studied, including biogas performance, process stability, substrate degradability, enzyme activity, and microbial communities. The results showed that the removal rates of erythromycin for all tested concentrations were higher than 90% after fermentation. Erythromycin addition inhibited biogas production. The more erythromycin added, the lower the CH_4_ content obtained. The high concentration of erythromycin (20 and 40 mg/L) resulted in more remarkable variations of pH values than the control group and 1 mg/L erythromycin added during the fermentation process. Erythromycin inhibited the hydrolysis process in the early stage of anaerobic fermentation. The contents of chemical oxygen demand (COD), NH_4_^+^–N, and volatile fatty acids (VFA) of erythromycin added groups were lower than those of the control group. Erythromycin inhibited the degradation of lignocellulose in the late stage of fermentation. Cellulase activity increased first and then decreased during the fermentation and addition of erythromycin delayed the peak of cellulase activity. The inhibitory effect of erythromycin on the activity of coenzyme F_420_ increased with elevated erythromycin concentrations. The relative abundance of archaea in erythromycin added groups was lower than the control group. The decrease in archaea resulted in the delay of the daily biogas peak. Additionally, the degradation rate of erythromycin was significantly correlated with the cumulative biogas yield, COD, pH, and ORP. This study supports the reutilization of antibiotic-contaminated biowaste and provides references for further research.

## 1. Introduction

Antibiotics are a particularly important type of drug and are widely used in human and veterinary medicine, which can prevent or treat diseases, improve feed efficiency and the growth rate of livestock [1]. However, antibiotics are difficult to be absorbed by domestic animals and most of them are not completely metabolized. Moreover, 70–90% of antibiotics would be eliminated in the original form through feces and urine [2]. As a promising technology to produce bioenergy, anaerobic fermentation not only treats organic waste, but also recycles renewable energy [3]. Anaerobic fermentation was used as an effective means to remove antibiotic resistance genes and pathogenic microorganisms [4,5]. It is of great significance to study the mechanism of antibiotic removal based on anaerobic fermentation. 

The influence mechanism of antibiotics on anaerobic fermentation is complex. It was found that the antibiotics of normal range concentration made no difference in the CH_4_ process in the pig manure fermentation [6]. However, Loftin et al. [7] suggested that even very low concentrations of antibiotics reduced the CH_4_ yield. After adding chlortetracycline into the anaerobic fermentation system of pig manure, the daily gas yield of the experimental group in the first 10 days was significantly lower than the control group [8]. Feng et al. [9] studied the anaerobic fermentation of pig manure with a few different kinds of antibiotics. It was found that only erythromycin was degraded during anaerobic digestion. Aydin et al. [10] investigated impacts of mixed antibiotics on COD removal, VFA production, antibiotic degradation, biogas production and composition, and found that erythromycin can have an antagonistic effect on sulfamethoxazole and tetracycline. 

Erythromycin and its derivatives are the most widely used macrolide antibiotics and anti-infective agents [11]. Improper disposal of erythromycin not only led to the dissemination of antibiotic resistance genes (ARGs), but also affected their underlying host microbial communities [12]. However, only few studies have examined erythromycin degradation during anaerobic fermentation. Alenzi et al. [13] suggested that anaerobic microorganisms provided a means of enhanced removal of erythromycin, which was removed completely during the long-term digestion of synthetic sewage sludge. Wang et al. [14] reported that 80% of erythromycin in the erythromycin fermentation dregs was removed after 30 days of mesophilic digestion. Moreover, a high concentration of erythromycin (500 mg/L) increased by 13% rather than inhibited the methane yields [14]. Despite the abovementioned research, the effect mechanism of erythromycin on anaerobic fermentation with feces as feedstocks remains unclear, particularly the relationships between microorganisms and biodegradability.

The objective of this research is to reveal the effect mechanism of erythromycin on anaerobic fermentation by studying the biodegradability and microbial characteristics under erythromycin stress. The removal of erythromycin, the effect of erythromycin on biogas properties, process stability, substrate degradation, enzyme activity, as well as the microorganisms during the fermentation process were investigated. This study is expected to provide a reference for the control and removal of erythromycin.

## 2. Materials and Methods

### 2.1. Experimental Materials

The feedstocks included corn stover and cow dung. The corn stover was collected from the farmland in Zhangjiakou city, Hebei province, China, in November 2018. It was harvested by cutting 10 cm above the ground, then diced into pieces 5–10 cm in length and air-dried until moisture levels were below 10%. After drying, the corn stover was ground into powder and passed through a 10-mesh sieve. The cow dung was collected from Jin Yuan Husbandry Company (Zhangjiakou, China) in June 2019. The fresh cow dung was stored at 4 °C. Table 1 shows the characteristics of the corn stover and cow dung.

### 2.2. Anaerobic Fermentation Experiment

The experiments were performed in the anaerobic fermenters (total volume of 30 L, available volume of 20 L, YGF 300/30, Shanghai Yangge Biological Engineering Equipment Co., Ltd., Shanghai, China) for 31 days under 37.0 ± 1.0 °C (automatically controlled). The fermenters were cleaned and autoclaved before each experiment. The digester contents were thoroughly stirred by a three-layer stirrer introduced in the middle of each fermenter for 30 min from 9:30 am every day.

Total solid (TS) of the substrate in fermenters was adjusted to 8% by adding distilled water. At the beginning of fermentation, the concentrations of erythromycin were set to 1, 20, and 40 mg/L through dissolving into a small amount of sodium hydroxide solution. No erythromycin was added to the control group. Finally, the fermenters were purged with N_2_ gas for 10 min to remove the oxygen. The experimental flow chart performed by CmapTools (as recommended by Behzadi et al. [15]) is shown in Figure 1.

### 2.3. Chemical and Microbial Analyses

The top outlet of the fermenter was connected to a gas volume detection device and the daily biogas yield was determined at 10:00 am every day, indicating the gas volume produced during the last 24 h. The pH value of the sample was determined by Rex pH meter (PHS-3C, Shanghai INESA & Scientific Instrument Co., Ltd., Shanghai, China). The ORP values were automatically measured by an ORP electrode (OxyTrode Pt 120 P/N: 233810, Hamilton Company, Bonaduz, Switzerland) connected to the fermenter. The biogas in the tank was collected at 10:00 am every 3 days by the gas sampling bag. Liquid and solid samples were collected by the sampling port at the bottom of the reactor every 3 days. TS was measured after drying at 105 °C for 24 h. Volatile solids (VSs) were measured after treating the samples in a muffle furnace at 550 °C for 1 h. Ammonia nitrogen (NH_4_^+^–N) was measured by Nessler’s reagent method [16]. COD in the supernatant was obtained by the potassium dichromate method after sample centrifugation at 5000 rpm for 10 min [17]. Samples for VFA analysis were passed through a 0.45 μm nitrocellulose membrane filter and frozen before analysis. VFA were measured using a gas chromatograph (GC–2014, Shimadzu Co., Kyoto, Japan) equipped with free fatty acid phase (FFAP) column with a flame ionization detector (FID). VFAs were expressed as mg/L of individual species (C2–C5 fatty acids). Cellulose, hemicellulose, and lignin in solid were determined by Enzyme-Linked Immunosorbent Assays kit (ELISA, Qingdao Kebiao Testing and Research Institute Co., Ltd., Qingdao, China). The cellulase and coenzyme F_420_ activities in the supernatant were determined, in accordance with the standard method after centrifugation at 4000 rpm for 5 min [18]. The concentration of erythromycin was determined by a liquid chromatography-tandem mass spectrometry equipped with an Agilent Eclipse C18 column (Agilent G6460A, USA). CH_4_ contents in biogas were measured by a gas chromatograph (GC–2014C, Shimadzu Co., Japan) equipped with a GDX–401 column with H_2_ as the carrier gas. Detection was performed with a thermal conductivity detector (TCD). The concentrations of metal elements and S element were measured by atomic absorption spectrophotometry (Pony Testing International Group Co., Ltd., Hefei, China). The coliform bacteria and mortality rate of ascaris eggs were determined by Pony Testing International Group Co., Ltd. (GB/T 19524.1-2004, GB/T 19524.2-2004).

The measurement of microbial communities was conducted by the Beijing Ovation Genetics (Beijing, China) after certificating the samples at 8000 rpm and 4 °C for 3 min. By directly amplifying the specific region of total DNA in the sample, the distribution and abundance in samples can be effectively evaluated. The genomic DNA of each sample was extracted by the Cetyltrimethylammonium Ammonium Bromide (CTAB) method [19]. Bacterial V3–V4 region gene amplification was performed using universal primers 338F (5′-ACTCCTACGGGAGGCAGCAG-3′) and 806R (5′-GGACTACHVGGGTWTCTAAT-3′). Archaea V3-V4 region gene amplification was performed using universal primers 344F (5’-ACGGGGYGCAGCAGGCGCGA-3′) and 806R (5’-GGACTACVSGGGTAT CTAAT-3′). Three parallel tests were performed on each sample. The PCR products were detected by agar gel electrophoresis and sequenced by 16S rDNA high throughput sequencing technology. At the 97% similarity level, the RDP algorithm was used to analyze the representative sequences of operational taxonomic units (OTUs) and the specific information of the community was annotated at each level. On this basis, alpha diversity, beta diversity, and relative abundance distribution at the level of phylum and genus were calculated.

### 2.4. Data Analysis

The data in the study were the average of triplicate treatments. Pearson correlation analysis was used to analyze the fluctuation of antibiotic concentration and the correlation between antibiotic degradation rate and fermentation parameters. The relationships between antibiotic concentration fluctuation, antibiotic degradation rate, and microorganisms were also analyzed. Pearson correlation analysis was performed in Statistical Package for the Social Science (IBM SPSS Statistics, 23,IBM Co., New York, NY, USA) software at 0.05 and 0.01 levels of significance by * (*p* < 0.05) and ** (*p* < 0.01), respectively.

## 3. Results and Discussion

### 3.1. Erythromycin Removal

As shown in Figure 2A, erythromycin was effectively removed during the fermentation process. The removal rate of erythromycin was more than 90% in all the groups, which was higher than the previous study [14]. During the whole anaerobic fermentation process, the concentration of erythromycin was decreased continually. The removal efficiency of erythromycin increased first and then decreased with the highest efficiency on the 7th day (Figure 2B). These results agreed with a previous study that found the degradation rate of erythromycin was higher in the early stage of fermentation and it could be up to 76.60% during the first 5 days [20]. It was found that the degradation rate of erythromycin decreased with the extension of time and reached the degradation rate of 94.44% on the 10th day [21]. The results of this study matched with those of previous studies.

### 3.2. Impacts of Erythromycin on Biogas Properties

#### 3.2.1. Biogas Yields

Figure 3A shows the impact of adding erythromycin on cumulative biogas yields. The higher the initial concentration of erythromycin, the less the cumulative biogas yields of the anaerobic fermentation system. The cumulative biogas yields of the erythromycin added groups with added concentrations of 1, 20, and 40 mg/L were 10.31%, 33.65%, and 34.87% lower than the control group, respectively. 

In terms of the dynamic fluctuation of the cumulative biogas yields throughout the process, the biogas yields trend of the 1 mg/L erythromycin added group was close to the control group and the biogas yields of the 20 mg/L erythromycin added group was close to the 40 mg/L erythromycin added group. For the control group, cumulative biogas yields were 477.48 mL/g TS, which was considerably higher than the 20 and 40 mg/L erythromycin added groups. The cumulative biogas yields of the 1, 20, and 40 mg/L erythromycin added groups were 89.69%, 66.35%, and 65.13% of the control group. In the control and 1 mg/L groups, the cumulative biogas yields increased sharply from the 10th day of fermentation, while in the 20 and 40 mg/L erythromycin added groups, the cumulative biogas yields increased slightly on the 12th day of fermentation. Therefore, erythromycin not only inhibited the biogas production, but also delayed the peak of biogas yields. 

During the first 15 days of anaerobic fermentation, the biogas yields of the control and 1 mg/L groups were significantly higher than the 20 and 40 mg/L erythromycin added groups, demonstrating that erythromycin reduced biogas yields during the start-up period of anaerobic fermentation. The highest daily biogas yields were 32.59, 27.00, 18.54, and 19.40 mL/g TS for the control as well as 1, 20, and 40 mg/L for the erythromycin added groups. The second peak of the four groups was reached on the 21st, 22nd, and 24th day, respectively. The daily biogas yields were 20.8, 18.5, 16.3, and 17.0 mL/g TS, respectively. Compared with the control group, the second peak biogas yields decreased by 10.91%, 21.63%, and 18.27% for the 1, 20, and 40 mg/L erythromycin added groups, respectively. These results show that erythromycin reduced biogas yields during the late-stage of anaerobic fermentation, and delayed the emergence of the second peak. In conclusion, erythromycin addition inhibited the hydrolysis process, prolonged the start-up period of anaerobic fermentation, and finally reduced the total biogas yields. The inhibitory concentration in the present study was considerably lower than the previous research of Wang et al. (2022), in which 500 mg/L of erythromycin induced another small peak at the 22nd day, indicating that the system recovered from the high antibiotic pressure. On the one hand, the different feedstocks might influence the renitency to erythromycin stress. On the other hand, they might impact the ultra-high concentration of erythromycin on the fermentation process, particularly the late stage of fermentation, which requires further exploration.

#### 3.2.2. CH_4_ Yields

The effect of erythromycin on CH_4_ contents is shown in Figure 3C. On the 1st day of anaerobic fermentation, CH_4_ was detected in all four groups of experiments. At this time, the stage of acid production and CH_4_ production had started. The CH_4_ contents on the 1st day of the four groups were 7.6%, 8.0%, 6.0%, and 6.4%, respectively. Therefore, the CH_4_ production during the start-up period was promoted slightly in the 1 mg/L added group.

During the first 7 days of anaerobic fermentation, the lower CH_4_ contents in the three erythromycin added groups indicated that erythromycin inhibited the biogas production by inhibiting the initiation of anaerobic fermentation. From the 7th to 16th day of anaerobic fermentation, the CH_4_ contents of the four groups showed a rapidly rising trend, i.e., the methanogenic process was active during this time. After the 16th day of fermentation, the methanogenic process was constant. The highest CH_4_ contents were 65.7%, 63.0%, 49.6%, and 50.0% for the control and 1, 20, and 40 mg/L for the erythromycin added groups, respectively. High concentration of erythromycin (above 20 mg/L) inhibited CH_4_ production remarkably. The inhibitory effect of erythromycin on methanogens was previously studied by Liu et al. [22], who found that by adding 20 mg/L of erythromycin into the fermentation system, CH_4_ content decreased from 68% to 49%. 

### 3.3. Impact of Erythromycin on Process Stability

#### 3.3.1. The pH Values

The variations of pH values during the fermentation process are shown in Figure 4A. In general, the pH values of all groups decreased at the beginning and then recovered. This was due to the fact that the organic components were quickly hydrolyzed into acids during the start-up stage of fermentation. Therefore, acidic hydrolytic products accumulated, resulting in a decrease in pH values [23]. The minimum pH values of the four groups were 6.36, 6.31, 6.01, and 5.88, respectively. The pH values of the 20 and 40 mg/L erythromycin added groups were lower than the control group remarkably. This was due to the fact that the high concentration of erythromycin inhibited the methanogenic process and the acid accumulated for a longer period of time in the system.

After the 7th day of anaerobic fermentation, the pH values of the control and 1 mg/L groups increased continuously. The pH values of the 20 and 40 mg/L erythromycin added groups increased from the 10th day. This proved that the methanogenic process was more active than the hydrolysis process. On the last day of anaerobic fermentation, the pH values of the four groups were 7.15, 7.04, 6.76, and 6.87, respectively. This corresponded to the previous reported pH range of 6.7–7.4 for the function of most methanogenic bacteria [24], as well as the pH range of 6.8–7.2 in the presence of mixed antibiotics [10].

#### 3.3.2. ORP

The ORP values represent the oxidation reduction reactions in the fermenters, and the lower the ORP value, the deeper the anaerobic level of the fermentation system [25]. The effects of erythromycin on the ORP values are rarely reported. As shown in Figure 4B, the ORP values of all four groups showed a downward trend. On the 1st day of fermentation, the ORP values of the four groups were −131, −121, −97, and −87 mV, respectively. Therefore, the ORP values in the erythromycin added groups were higher than the control group and increased with the increase in erythromycin concentration. From the 10th to 15th day, the ORP value of the control group increased in fluctuation. Thereafter, the ORP values of the control and 1 mg/L groups remained constant. The ORP values of the 20 and 40 mg/L erythromycin added groups were higher than the control group during the whole process and remained constant after the 24th day. The results corresponded to the previous study that lower ORP values were beneficial for the CH_4_ production and biogas accumulation [26].

### 3.4. Substrate Biodegradation

#### 3.4.1. Variations of NH_4_^+^–N Concentrations

NH_4_^+^–N was an intermediate product of the anaerobic fermentation process. It improved the buffer performance of the system for volatile organic acids [18]. During the whole fermentation process, the concentrations of NH_4_^+^–N in all groups were increased first and then decreased (Figure 5). Compared with the control group, the NH_4_^+^–N concentration of erythromycin added groups fluctuated considerably. The results indicated that the stability of the NH_4_^+^–N concentration was weakened under erythromycin stress. During the first 5 days of fermentation, the NH_4_^+^–N concentrations of 20 and 40 mg/L erythromycin added groups decreased sharply. It was speculated that high erythromycin inhibited the NH_4_^+^–N generation process in the hydrolysis process. 

NH_4_^+^–N was reported to promote the CH_4_ production when the concentration was <6000 mg N/L [27]. In general, the NH_4_^+^–N concentrations of all groups were maintained below 450 mg/L. The average concentrations of the four groups were 360, 357, 307, and 319 mg/L, respectively. Therefore, adding erythromycin reduced the NH_4_^+^–N concentration and nitrogen source for microorganisms.

#### 3.4.2. Responses of VFAs

VFAs were further utilized by microorganisms to produce CH_4_ and other products. This was an important index to evaluate the balance of hydrolysis, acidification, acetogenesis, and methanogenesis stages [28]. Figure 6 shows the VFA concentrations of the four groups. In general, the total VFA concentrations of all groups increased continuously during the start-up stage of the fermentation. Meanwhile, the corresponding pH value decreased rapidly (Figure 4A). The organic materials in the substrate were quickly hydrolyzed into acids. Due to the low efficiency of methanogenesis, the acidic hydrolytic products accumulated, resulting in a decrease in pH values [29]. During this stage (the first 10 days), the total VFA concentrations of the 20 and 40 mg/L erythromycin added groups were lower than the control group considerably. This suggested that the high concentration of erythromycin addition inhibited the acidification process, which affected the subsequent methanogenic process and resulted in low biogas yields. From the 10th to 13th day of fermentation, the total VFA concentration of the control group decreased remarkably, while the total VFA concentrations of the 20 and 40 mg/L erythromycin added groups increased. The results confirmed that the high concentration of erythromycin addition (more than 20 mg/L) delayed the anaerobic fermentation process. After the 25th day of anaerobic fermentation, the total VFA concentrations of all groups decreased to below 700 mg/L. At this time, the biogas yields of all groups decreased to the lowest level, and the fermentation nearly stopped.

The concentrations of VFA components varied with the erythromycin concentrations. In general, acetic acid and butyric acid were the main components. During the whole fermentation process, acetic acid increased first and then decreased. Acetic acid concentrations of the 20 and 40 mg/L erythromycin added groups were remarkably lower than the control and 1 mg/L groups. Therefore, the biogas and CH_4_ yields of the 20 and 40 mg/L erythromycin added groups were low. Moreover, the low efficiency of transferring C3–C5 VFAs into acetic acid limited biogas production. After the 13th day (10th day for the control group), the concentrations of acetic acid in all groups decreased. This indicated that acetic acid was efficiently used during methanogenesis. Meanwhile, the CH_4_ yields of all groups increased rapidly (Figure 3C). 

#### 3.4.3. COD

COD represented the content of organic matter in the system. This was a comprehensive reflection of the hydrolysis, acidification, acetogenesis, and methanogenesis stages in the fermentation process [17]. During the whole fermentation process, the tendencies of the four groups showed similar performance (Figure 7). On the 1st day of fermentation, the initial COD of the control, 1, 20, and 40 mg/L erythromycin added groups were 6051, 5797, 5732, and 5647 mg/L, respectively. This proved that erythromycin reduced the organic matter in the system.

The COD of the control group reached peak value on the 7th day and the COD of the three erythromycin added groups reached the peak on the 10th day. Considering the peak time of the daily biogas yields (Figure 2B), the results indicated that organic components in the liquid phase were utilized for biogas production. COD concentrations decreased further until the 19th day. However, the biogas yields remained at a low level. It is possible that the COD contents in the substrate cannot be directly used by the methanogens. From the 19th to 25th day of anaerobic fermentation, the COD of the control and 1 mg/L groups showed an upward trend. The biogas yields of the four groups reached the second peak during this period. On the last day of fermentation, COD in all groups decreased to below 6000 mg/L. Considering the whole experimental process, the degradation of organic matter was delayed in the erythromycin added groups together with the delayed biogas production.

#### 3.4.4. Degradation of Lignocelluloses

In this study, both the cow dung and corn straw were rich in lignocellulose, i.e., hemicellulose, cellulose, and lignin (Table 1). Cellulose and hemicellulose were easy to degrade when they exist alone in the environment. However, their compounds were difficult to degrade by microorganisms [30]. The average contents of lignocellulose (sum of cellulose, hemicellulose, and lignin) in the four groups were 44.54, 52.37, 49.42, and 48.51 % TS, respectively (Figure 8). Erythromycin inhibited the degradation of lignocellulose.

The contents of lignocellulose in all groups remained constant during the first 19 days of fermentation, indicating that lignocellulose was difficult to degrade. The easily degradable organic matter in the hydrolysis process mainly came from cattle manure. On the 19th day of anaerobic fermentation, the contents of lignocellulose in the control, 1, and 20 mg/L erythromycin added groups decreased. Meanwhile, the daily biogas yields of the three groups reached the second peak. On the 22nd day of fermentation, the lignocellulose content of the 40 mg/L erythromycin added group decreased together with the appearance of the second daily biogas yield peak. It was speculated that the lignocellulose in the fermentation system was hydrolyzed to acid and small molecular compounds in the middle and late stages (day 19–22). Small molecular compounds provided a precursor for the methanogenic process, resulting in the second daily biogas yield peak. After the 22nd day, the contents of lignocellulose in all groups remained constant.

The effects of erythromycin addition varied with the compositions of lignocellulose (Table 2). On the one hand, the high concentration of erythromycin (≥20 mg/L) into fermenters inhibited the degradation of cellulose significantly (one-way ANOVA, *p* < 0.05) and reduced the hydrolytic products for further fermentation, resulting in the low biogas yields. On the other hand, erythromycin addition did not inhibit the degradability of hemicellulose and cellulose (*p* > 0.05). Nevertheless, the mechanisms require further study in the future.

### 3.5. Responses of Enzyme Activity

#### 3.5.1. Cellulase

Hydrolysis is the rate-limiting step of anaerobic fermentation [31]. The extracellular enzymes, such as cellulases and lipases, have been shown to be effective in the hydrolysis process [32]. Cellulase transformed lignocellulose into soluble sugar during fermentation [33]. Generally, the cellulase activities of all groups increased first and then decreased (Figure 9A). The average cellulase activities of the four groups were 48.98 ± 2.18, 49.39 ± 2.09, 46.03 ± 1.92, and 42.07 ± 1.66 μg/mL min, respectively. On the 1st day of anaerobic fermentation, the cellulase activities of the control and 1 mg/L groups were remarkably higher than the other groups. The high concentration (above 20 mg/L) of erythromycin inhibited the cellulase activity during the start-up stage of fermentation. However, the cellulose concentrations of the four groups were almost unchanged during the first 16 days. This was possibly due to the fact that the degradation of hemicellulose and other non-lignocellulose components might result in the percentage of cellulose in the total solid (TS).

The cellulase activities of the four groups reached the maximum value of 66.29, 63.47, 58.34, and 50.11 μg/mL min on the 16th, 19th, 19th, and 22nd day, respectively. The cellulose concentrations of the four groups decreased from the 19th day (22nd day for the 40 mg/L group). Meanwhile, the four groups reached the second biogas yields peak from the 21st to 24th day, respectively. The degradation of the cellulose was hypothesized to be the reason for the emergence of the second peak.

#### 3.5.2. Coenzyme F_420_

The activity of coenzyme F_420_ represented the activity of methanogens and can be used for monitoring the activity of methanogens [34]. Variations of coenzyme F_420_ activities are shown in Figure 9. During the whole anaerobic fermentation process, the activities of coenzyme F_420_ in the four groups showed a trend of decreasing first and then increasing. The average activities of coenzyme F_420_ in the four groups were ranked from large to small as the control, 1, 20, and 40 mg/L groups, respectively. It showed that the higher the concentration of erythromycin, the greater the inhibition of coenzyme F_420_ activity.

From the 4th to 19th day, with a small increase in the activities of coenzyme F_420_ in the four groups, the CH_4_ yields increased rapidly (Figure 3C). The results indicated that there might be other reasons to explain the high concentration of CH_4_. Methanogens using H_2_/CO_2_ as the matrix usually contain two hydrogenases: One is hydrogenase, which uses coenzyme F_420_ as the electron acceptor, so-called coenzyme F_420_-reducing hydrogenase. The other is the coenzyme F_420_ non-reducing hydrogenase [35]. Therefore, the coenzyme F_420_ functions in the H_2_/CO_2_ pathway for CH_4_ production. However, about 70% of CH_4_ is generated from acetic acid [36]. As a result, the final CH_4_ yields may not show the same trend as the coenzyme F_420_ activity.

### 3.6. Impacts of Erythromycin on the Microbial Communities

#### 3.6.1. Bacterial Communities

In this study, samples were collected on the 4th, 10th, 16th, 22nd, and 28th day, respectively. The samples of the four groups were marked as K1, K2, K3, K4, K5, E1_1, E1_2, E1_3, E1_4, E1_5, E2_1, E2_2, E2_3, E2_4, E2_5, E3_1, E3_2, E3_3, E3_4, and E3_5, respectively. 

Figure 10A shows the variations of bacterial communities annotated on the level of the genus. In the control group, *Bacteroides*, *Prevotella_7*, *Caproiciproducens*, *Escherichia-Shigella*, *Runminofilibacter*, and *Ruminiclostridium* were the dominant bacteria groups, with relative abundance ranging from 32.9% to 56.3%, 4.0% to 12.5%, 1.3% to 7.5%, 0.5% to 4.3%, 0.5% to 4.1%, and 0.5% to 5.0%, respectively. *Bacteroides* species are normally mutualistic, making up the most substantial portion of the mammalian gastrointestinal microbiota [37]. They can use simple sugars when available [38] and convert amino acids, sugars, and alcohols into VFAs [39]. The relative abundance of *Bacteroides* in the control group was considerably higher than the three erythromycin added groups. This was hypothesized to inhibit growth by erythromycin. The results were consistent with the previous studies, in which erythromycin prevents bacterial growth [10]. *Prevotella*, previously classified in the genus *Bacteroides*, was an obligate anaerobic Gram-negative rod-shape bacterium [40]. Although it generally had a limited ability to ferment amino acid, *Prevotella* was also significantly correlated with multiple pathways involved in drug metabolism, carbohydrate metabolism, and metabolism of cofactors and vitamins, including vitamin B6 metabolism [41]. *Caproiciproducens* produced H_2_, CO_2_, ethanol, acetic acid, butyric acid, and caproic acid as metabolic end products of anaerobic fermentation [42]. The model microorganism *Ruminiclostridium* produced extracellular multi-enzymatic complexes known as cellulosomes, which efficiently degrade the crystalline cellulose [43]. The relative abundance of *Ruminiclostridium* on the 22nd day was increased in the 1 and 20 mg/L groups (E1_4, E2_4). Meanwhile, the biogas yields reached the second peak on the 22nd and 24th day, respectively.

The relative abundance of *Christensenellaceae_R-7_group*, *Rikenellaceae_RC9_gut_group*, *VadinBC27_wastewater-sludge_group,* and *Anaerotruncus* in erythromycin added groups was remarkably higher than the control group. *Christensenellaceae* belongs to the *Firmicutes* phylum of bacteria. *Christensenellaceae minuta* fermented glucose to acetate and butyrate under anaerobic conditions [44], which indicated that it fermented sugars in the gut to short-chain fatty acids and other fermentation products, such as H_2_ and CO_2_. To date, all cultured members of the family *Rikenellaceae* were described as anaerobic, mesophilic, and rod-shaped bacteria that usually ferment carbohydrates or proteins [45]. Therefore, the detected microbial communities in the erythromycin added groups supported the carbohydrates and proteins degradation. Limited information has been reported on the function of genus *vadinBC27*. It was reported as the most ruling genus detected in degrading the polybrominated diphenyl ethers (PBDE) with different electron donor amendments and its abundance significantly increased in the microcosms amended with electron donors [46]. In this study, genus *vadinBC27* might play a key role in the degradation of erythromycin. *Anaerotruncus* was catalase-negative, and produced acetic and butyric acids as end products of metabolism [47].

#### 3.6.2. Archaeal Communities

The variations of archaeal communities annotated on the level of the genus are shown in Figure 10B. Among all the samples, the most abundant archaea were *Methanobacterium*, *Methanobrevibacter*, *Methanosarcina*, *Methanosphaera*, *Methanofollis*, *Methanomassiliicoccus*, *Methanoculleus*, *Candidatus_Methanoplasma*, *Methanospirillum,* and *Methanogenium*. In this study, *Methanosarcina* was the dominant genus whose relative abundance was more than 20% in all groups.

Only *Methanosarcina* species possessed all three known pathways for methanogenesis and were capable of utilizing no less than nine methanogenic substrates, including acetate [48]. Its maximum relative abundance of the control group was higher than the three erythromycin added groups. This possibly explained the low CH_4_ yields of the erythromycin added groups. In contrast, Wang et al. [14] found that the relative abundance of *Methanosarcina* was enhanced in the presence of high concentrations of erythromycin. *Methanobacterium* is a common methanogen that utilizes acetic acid for methanogenesis in mesophilic anaerobic digestion [49]. *Methanobrevibacter* is strictly an anaerobic archaea, which produces methane mostly by the hydrogenotrophic pathway. For it to grow, ammonia is required as the nitrogen source and acetate as the main carbon source [50]. In general, the relative abundance of *Methanobrevibacter* in the four groups showed a decreasing trend. However, the concentration of NH_4_^+^−N did not show the same trend. In this case, it was speculated that there were other nitrogen sources related to this microorganism, which require further study.

### 3.7. Relationship Analysis between the Erythromycin Removal and Fermentation Parameters

#### 3.7.1. Relationship with Microbial Community

The Pearson correlation between the change in erythromycin content, the degradation rate, and the bacterial community are shown in Figure 11. There was a significant positive correlation between the content of erythromycin and the relative abundance of *Rikenellaceae_RC9_gut_group* (*p* < 0.05, R^2^ = 0.874; *p* < 0.01, R^2^ = 0.921; *p* < 0.01, R^2^ = 0.917 for 1, 20, and 40 mg/L erythromycin added groups, respectively). In the 1 and 20 mg/L erythromycin added groups, the changes in erythromycin content were positively correlated with the relative abundance of *Acidaminococcus* (*p* < 0.05, R^2^ = 0.854; *p* < 0.05, R^2^ = 0.832 for 1 and 20 mg/L erythromycin added groups, respectively). In the 20 and 40 mg/L erythromycin added groups, the contents of erythromycin were positively correlated with *Hydrogenophaga* (*p* < 0.05, R^2^ = 0.836; *p* < 0.01, R^2^ = 0.919 for 20 and 40 mg/L erythromycin added groups, respectively). The degradation rates of erythromycin were positively correlated with the relative abundance of *vadinBC27_wastewater-sludge_group* (*p* < 0.01, R^2^ = 0.981; *p* < 0.05, R^2^ = 0.875 for 20 and 40 mg/L erythromycin added groups, respectively). This confirmed the previous conjecture that *v**a**dinBC27_wastewater-sludge_group* played a key role in the degradation of erythromycin. In the 1 and 20 mg/L erythromycin added groups, the degradation rates of erythromycin were positively correlated with *Rikenellaceae_RC9_gut_group* (*p* < 0.05, R^2^ = 0.884; *p* < 0.01, R^2^ = 0.974 for 1 and 20 mg/L erythromycin added groups, respectively). 

The relative abundance of *Rikenellaceae_RC9_gut_group* and *vadinBC27_wastewater-sludge_group* in the three erythromycin added groups was considerably higher than the control group. Some studies showed that *Rikenellaceae_RC9_gut_group* was abundant in the rumen and played an important role in carbohydrate transport and metabolism [51]. Flint et al. [52] found that *Rikenellaceae_RC9_gut_group* played an important role in the degradation of non-cellulosic polysaccharides. It was speculated that the two kinds of bacteria were conducive to the degradation of erythromycin in the process of degradation of substrate.

Figure 12 shows the Pearson relationship between the major methanogens and erythromycin concentration (A) and erythromycin degradation rate (B). In the three erythromycin added groups, the erythromycin concentrations were positively correlated with the relative abundance of *Brevibacterium Methanatum* (*p* < 0.01, R^2^ = 0.985; *p* < 0.01, R^2^ = 0.973; *p* < 0.05, R^2^ = 0.887 for 1, 20, and 40 mg/L erythromycin added groups, respectively). In the 1 and 40 mg/L erythromycin added groups, the concentrations of erythromycin were positively correlated with *Methaspirillum* (*p* < 0.05, R^2^ = 0.891; *p* < 0.05, R^2^ = 0.869 for 1 and 40 mg/L erythromycin added groups, respectively). In the 20 and 40 mg/L erythromycin added groups, the concentrations of erythromycin were positively correlated with *Methanococcus* (*p* < 0.01, R^2^ = 0.992; *p* < 0.05, R^2^ = 0.876 for 20 and 40 mg/L erythromycin added groups, respectively). In the three erythromycin added groups, the degradation rates of erythromycin were positively correlated with *Methaspirillum* (*p* < 0.01, R^2^ = 0.993; *p* < 0.05, R^2^ = 0.721; *p* < 0.05, R^2^ = 0.843 for 1, 20, and 40 mg/L erythromycin added groups, respectively). In the 1 and 20 mg/L erythromycin added groups, the degradation rates of erythromycin were positively correlated with *Brevibacterium Methanatum* (*p* < 0.01, R^2^ = 0.982; *p* < 0.05, R^2^ = 0.989 for 1 and 20 mg/L erythromycin added groups, respectively). It was found that *Brevibacterium Methanatum* had good tolerance to the antibiotics, when the tolerance of microorganisms to antibiotics was studied in the anaerobic fermentation system [51]. It was suggested that *Brevibacterium Methanatum* was conducive to the erythromycin degradation in this study. However, the exact mechanism underlying the observed relationships between microbial abundance and erythromycin degradation still require further study.

#### 3.7.2. Relationship with Other Fermentation Parameters

The degradation of antibiotics was affected by many factors. Pearson correlation was used to analyze the correlation between antibiotics and anaerobic fermentation indexes (Figure 13). It can be seen from the figure that the degradation rate of erythromycin showed a significant negative correlation with the pH value (*p* < 0.01, R^2^ = −0.952; *p* < 0.01, R^2^ = −0.927 for 1 and 40 mg/L erythromycin added groups, respectively). A previous study reported a similar conclusion when the researchers studied the treatment of erythromycin residue by anaerobic fermentation. They found that the degradation efficiency of erythromycin was related to the pH value in the fermentation system [21]. The lower the pH value, the higher the degradation rate. In the 20 and 40 mg/L groups, the degradation rates of erythromycin were positively correlated with the COD value (*p* < 0.05, R^2^ = 0.773; *p* < 0.05, R^2^ = 0.804 for 20 and 40 mg/L erythromycin added groups, respectively). It was speculated that the degradation of erythromycin was more active in the early stage of anaerobic fermentation. The COD content increased rapidly and provided nutrients and energy for the growth of the microbial community, which improved the degradation efficiency of erythromycin.

There was a significant positive correlation between the concentration of erythromycin and ORP (*p* < 0.01, R^2^ = 0.957; *p* < 0.01, R^2^ = 0.982; *p* < 0.01, R^2^ = 0.931 for 1, 20, and 40 mg/L erythromycin added groups, respectively). In the three erythromycin added groups, the variations of erythromycin concentration were negatively correlated with the cumulative biogas yield (*p* < 0.05, R^2^ = −0.877; *p* < 0.01, R^2^ = −0.905; *p* < 0.01, R^2^ = −0.923 for 1, 20, and 40 mg/L erythromycin added groups, respectively) and the CH_4_ yield (*p* < 0.05, R^2^ = −0.864; *p* < 0.05, R^2^ = −0.889; *p* < 0.01, R^2^ = −0.954 for 1, 20, and 40 mg/L erythromycin added groups, respectively). The decrease in ORP and increase in the biogas and CH_4_ yields indicated that the anaerobic level of the fermentation system was deep. However, bacteria in the hydrolysis process were not strictly anaerobic. When the anaerobic level was deep, the activity of these bacteria, including those that degraded erythromycin, were affected.

### 3.8. Implications of This Study

High removal efficiency of erythromycin was achieved in the early stage of anaerobic fermentation. However, the inhibitory effect of erythromycin lasted for a longer period of time after degradation. On the one hand, although the tolerance of fermentation system to antibiotic stress was detected, the inhibitory concentrations of erythromycin should be assessed in accordance with the different feedstocks and fermentation conditions. On the other hand, the influence of metabolites should be taken into account. The anaerobic fermentation system is efficient in removing erythromycin and is an appropriate method for solving the problem of erythromycin pollution. Nevertheless, the use of antibiotics should be strictly reduced. Stricter standards should be set for antibiotic discharge.

## 4. Conclusions

In this study, we tried to reveal the mechanism of erythromycin removal and its effect on anaerobic fermentation by investigating the biogas performance, process stability, substrate degradability, enzyme activity, and microbial communities at different stages of fermentation. Anaerobic fermentation was effective in removing erythromycin (removal efficiency > 90%), and the removal process mainly occurred in the early stage of anaerobic fermentation. The addition of erythromycin reduced and delayed the biogas and CH_4_ yields. The inhibitory mechanism of erythromycin was demonstrated by deteriorations in process stability, low efficiency in transformation and utilization of VFAs, inhibition of enzyme activities, and limited degradation of lignin and cellulose. The added erythromycin reduced the relative abundance of *Bacteroides* and *Methanosarcina* remarkably, which was not conducive to the degradation of substrate and the production of CH_4_. Moreover, *Rikenellaceae_RC9_gut_group* and *vadinBC27_wastewater-sludge_group* were significantly correlated with the removal of erythromycin. In conclusion, it is feasible to remove erythromycin by anaerobic fermentation together with the biogas recovery. Nevertheless, the concentrations of erythromycin should be controlled in accordance with the feedstocks and reaction conditions.

## Figures and Tables

**Figure 1 ijerph-19-07256-f001:**
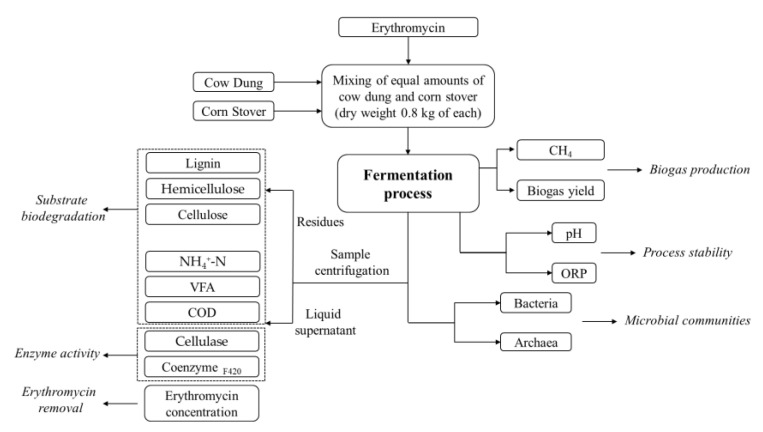
Experimental flow chart.

**Figure 2 ijerph-19-07256-f002:**
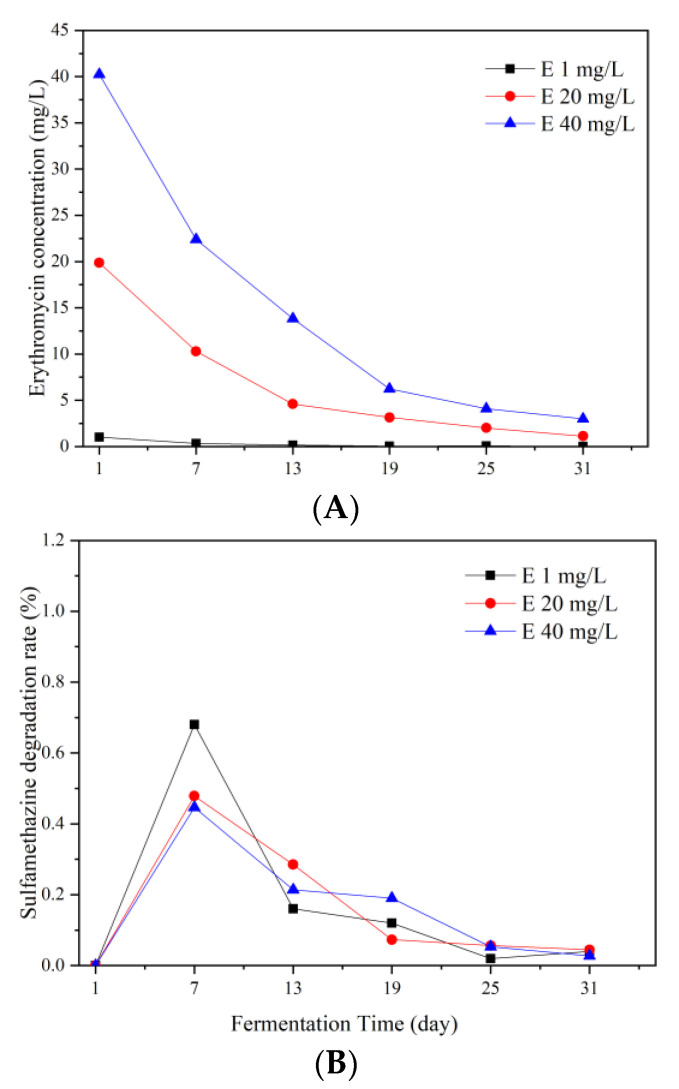
Variations of erythromycin concentration (**A**) and degradation rate of erythromycin (**B**) during the fermentation process.

**Figure 3 ijerph-19-07256-f003:**
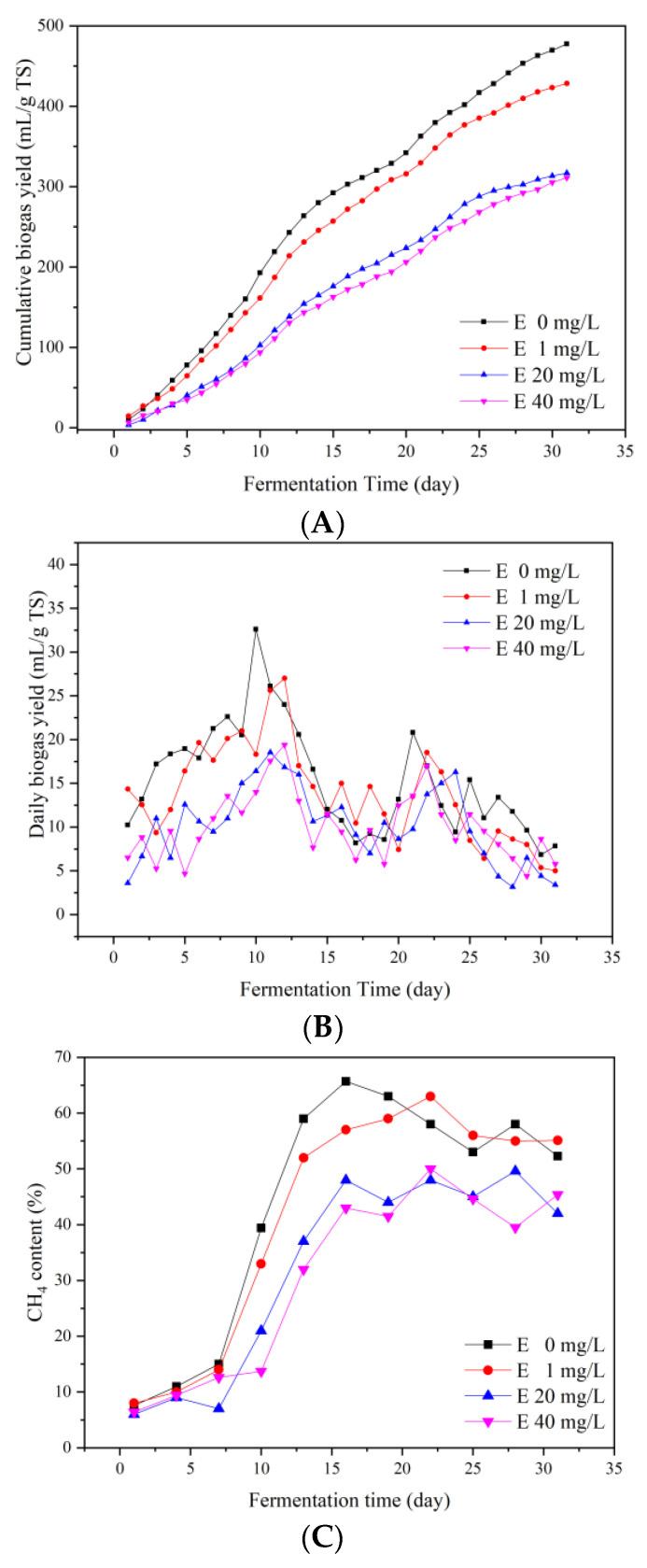
Cumulative biogas yields (**A**), daily biogas yields (**B**), and CH_4_ contents (**C**) during the fermentation process.

**Figure 4 ijerph-19-07256-f004:**
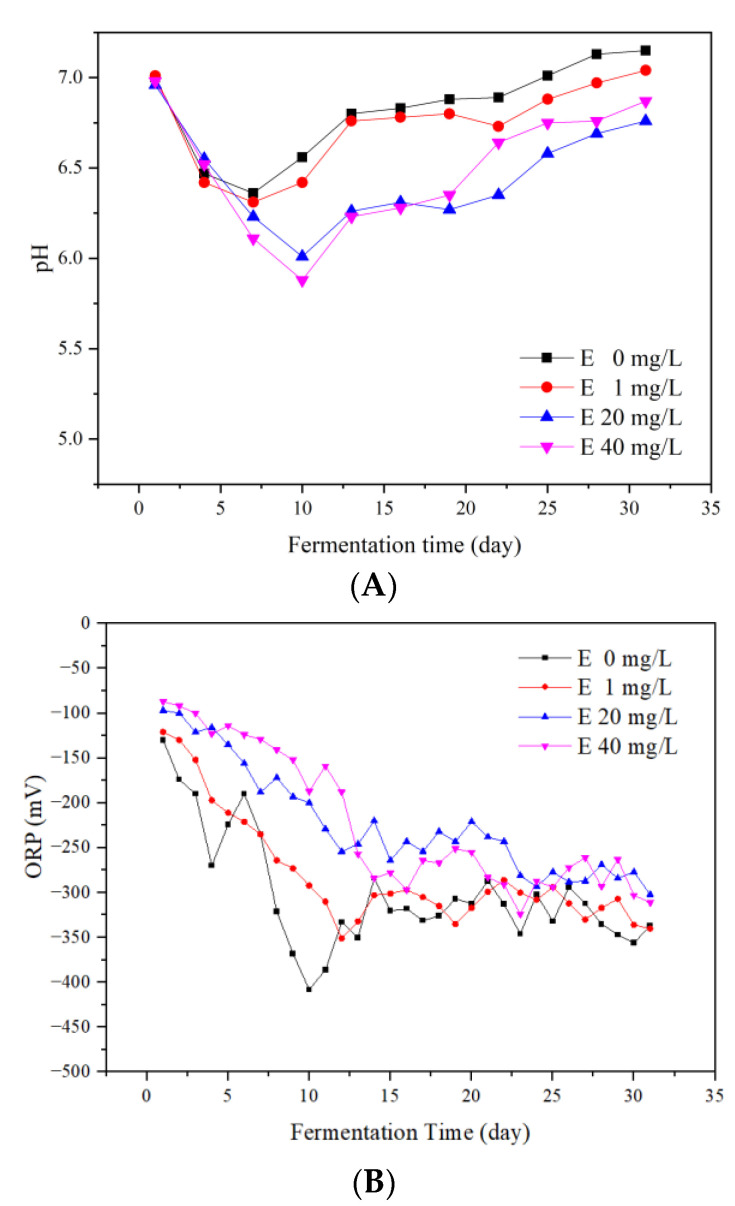
Variations of pH values (**A**) and ORP values (**B**) during the fermentation process.

**Figure 5 ijerph-19-07256-f005:**
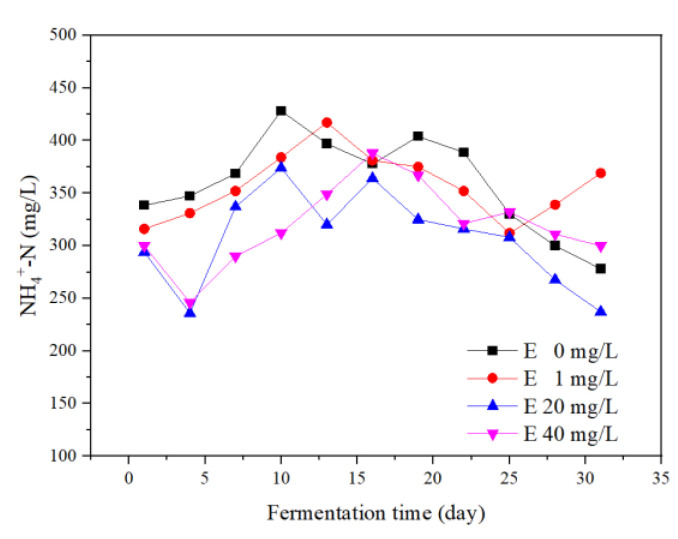
Ammonia nitrogen (NH_4_^+^–N) concentrations during the fermentation process.

**Figure 6 ijerph-19-07256-f006:**
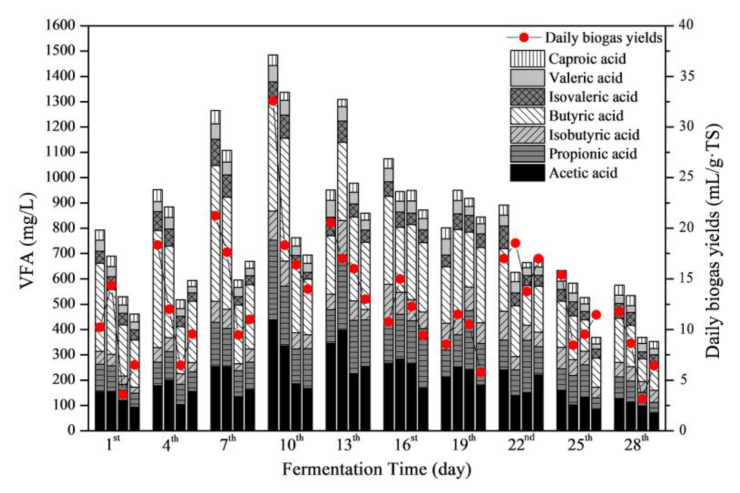
Volatile fatty acids (VFAs) concentrations in the liquid phase during the fermentation process. For each day, the column from left to right shows the VFA compositions of the control, 1, 20, and 40 mg/L erythromycin added groups, respectively. The compositions of VFA were shown by stacked bars with the order of acetic acid (black), propionic acid (transverse), isobutyric acid (bias), butyric acid (left bias), isovaleric acid (grid), valeric acid (gray), and caproic acid (vertical) from the bottom.

**Figure 7 ijerph-19-07256-f007:**
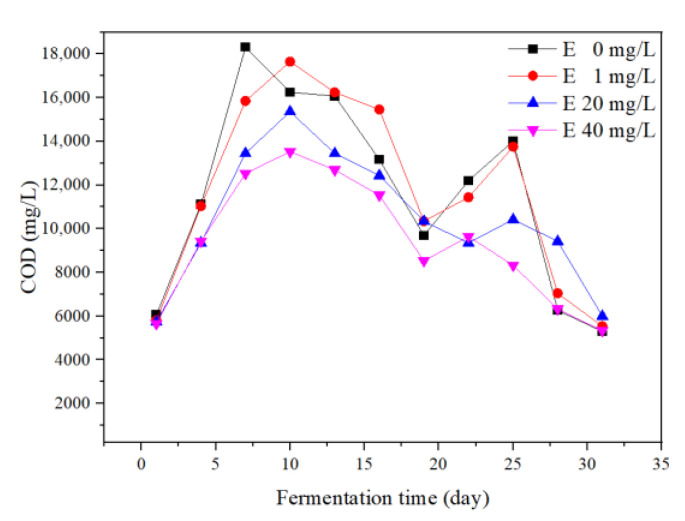
Chemical oxygen demands (COD) during the fermentation process.

**Figure 8 ijerph-19-07256-f008:**
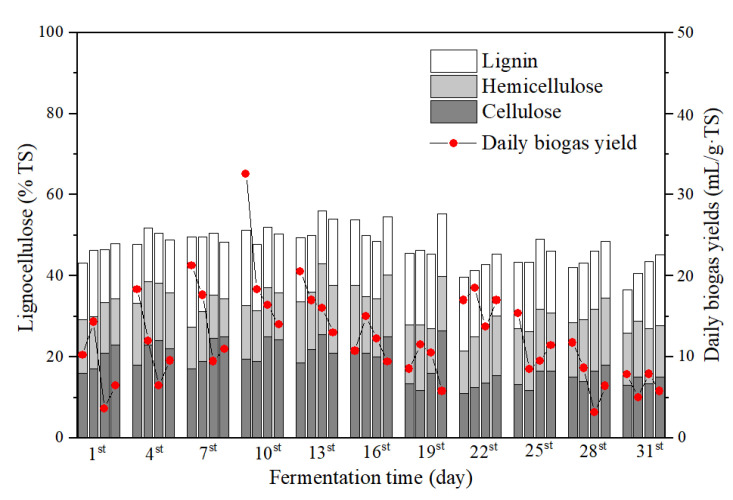
Lignocellulose contents during the fermentation process. For each day, the column from left to right shows the lignocellulose contents of the control, 1, 20, and 40 mg/L erythromycin added groups, respectively. The compositions of lignocelluloses were shown by stacked bars with the order of cellulose (dark gray), hemicellulose (gray), and lignin (blank) from the bottom.

**Figure 9 ijerph-19-07256-f009:**
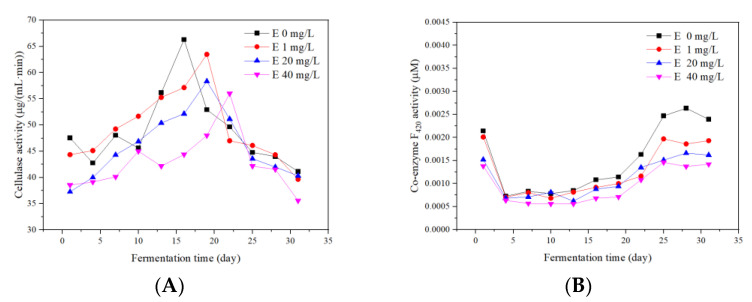
Responses of cellulase (**A**) and coenzyme F_420_ (**B**) during the fermentation process.

**Figure 10 ijerph-19-07256-f010:**
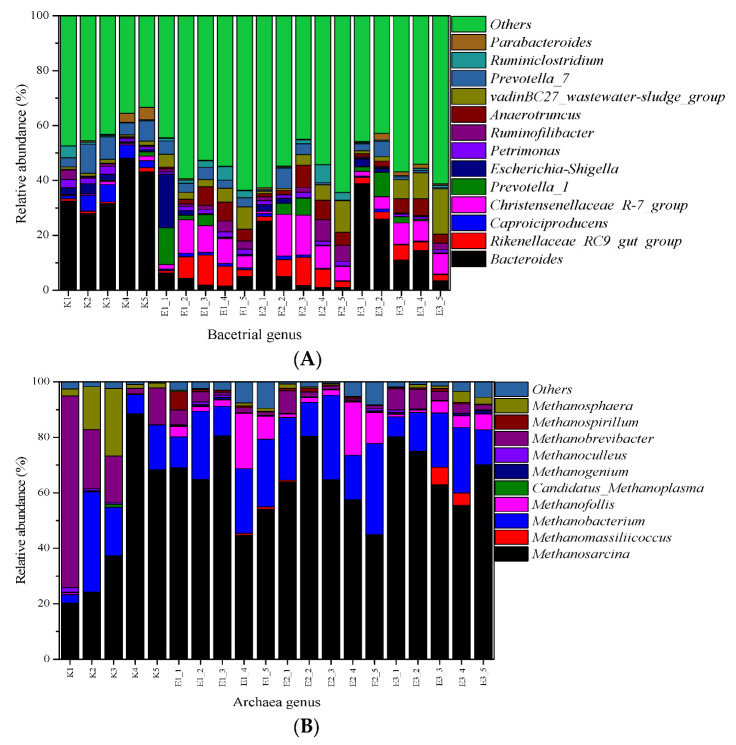
Bacterial communities (**A**) and archaeal communities (**B**) annotated on the level of the genus in control, 1, 20, and 40 mg/L erythromycin added groups on the 4th, 10th, 16th, 22nd, and 28th day of fermentation.

**Figure 11 ijerph-19-07256-f011:**
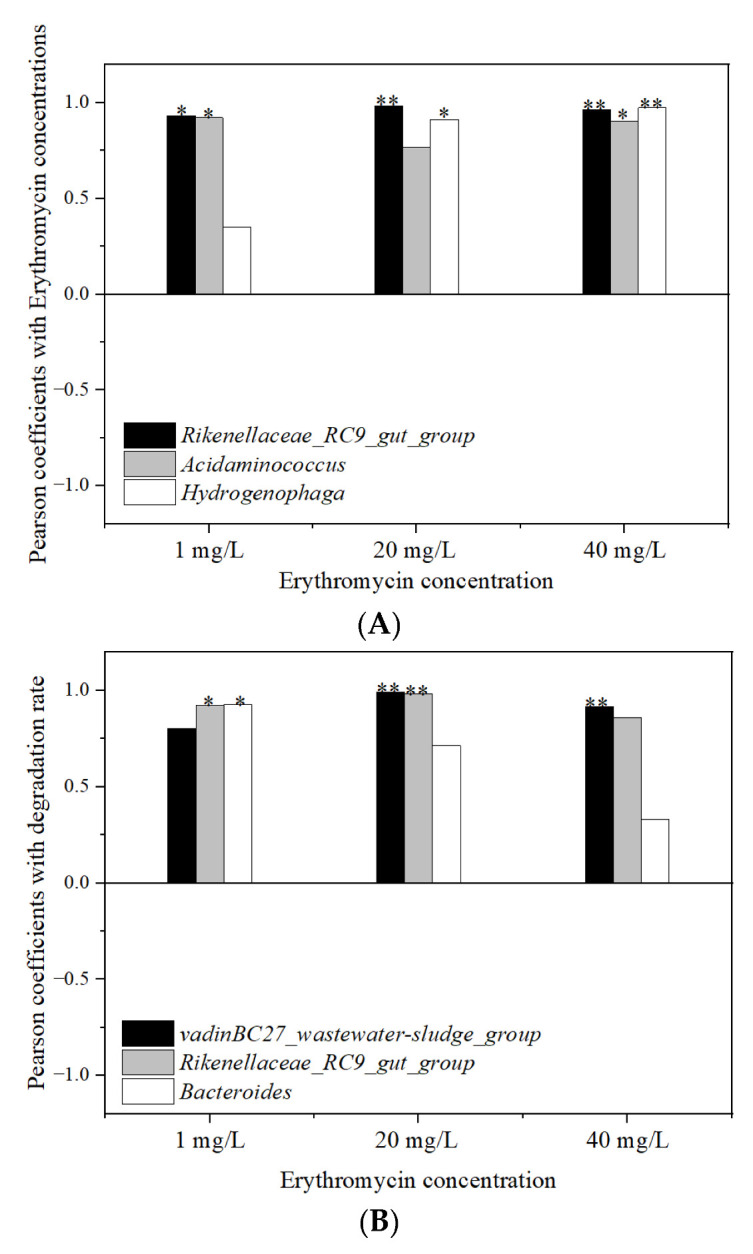
The correlation between the removal of erythromycin and the abundance of bacteria. (**A**) Pearson coefficient of erythromycin concentration and (**B**) Pearson coefficient of erythromycin degradation rate. * *p* < 0.05. ** *p* < 0.01.

**Figure 12 ijerph-19-07256-f012:**
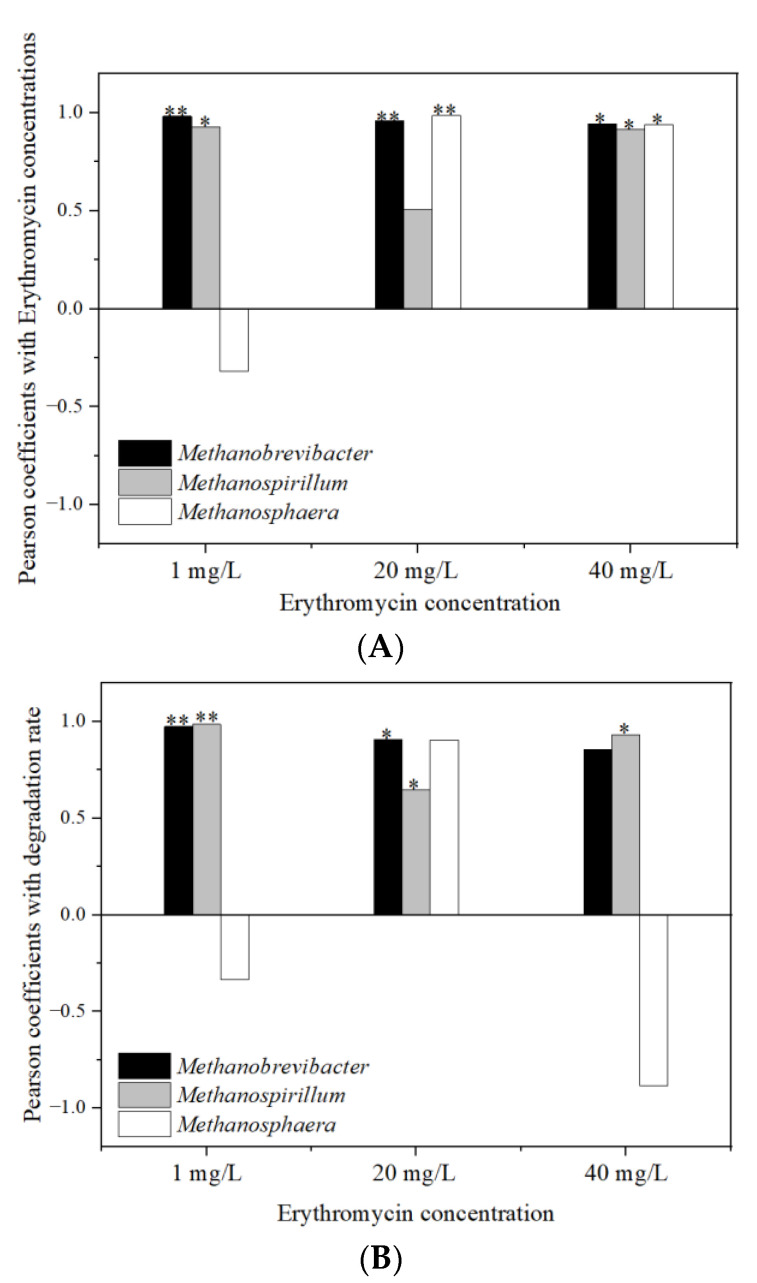
The correlation between the removal of erythromycin and the abundance of major methanogens. (**A**) Pearson coefficient between the abundance of major methanogens and erythromycin concentration and (**B**) Pearson coefficient between the abundance of major methanogens and erythromycin degradation rate. * *p* < 0.05. ** *p* < 0.01.

**Figure 13 ijerph-19-07256-f013:**
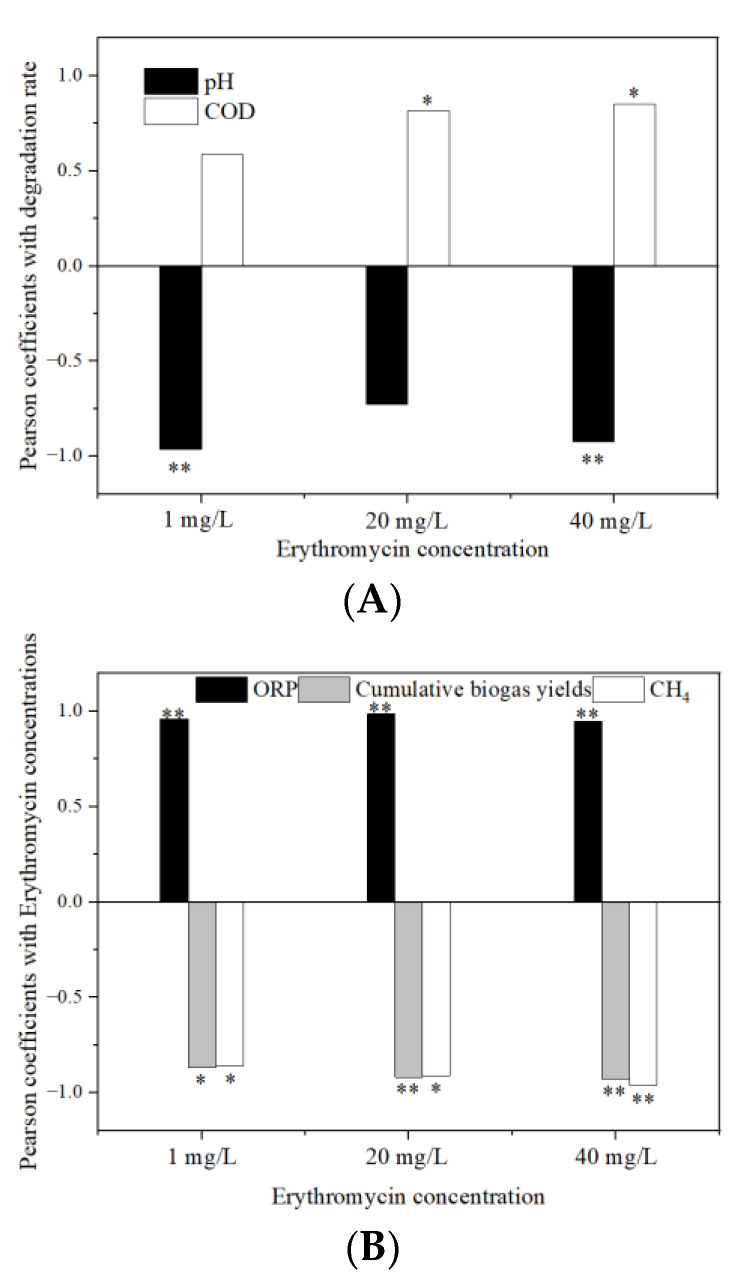
The correlation between the removal of erythromycin and fermentation parameters. (**A**) Pearson coefficient with erythromycin degradation rate and (**B**) Pearson coefficient with erythromycin concentration. * *p* < 0.05. ** *p* < 0.01.

**Table 1 ijerph-19-07256-t001:** Characteristics of corn stover and cow dung.

Characteristics	Cow Dung	Corn Stover
TS (% dry weight)	15.15 ± 0.41	91.24 ± 0.12
VS (% TS)	78.66 ± 0.06	86.34 ± 0.02
TN (% TS)	0.36 ± 0.01	0.34 ± 0.01
TOC (% TS)	25.42 ± 0.00	34.68 ± 0.00
Ratio of C/N	70.22	102.30
Cellulose (% TS)	26.92 ± 0.00	28.88 ± 0.00
Hemicellulose (% TS)	22.34 ± 0.00	24.92 ± 0.00
Lignin (% TS)	19.98 ± 0.00	21.30 ± 0.00
Mg (mg/kg)	8900.00	1400.00
Ca (mg/kg)	8000.00	1000.00
K (mg/kg)	1400.00	4200.00
Zn (mg/kg)	<5.00	7.10
Fe (mg/kg)	<5.00	170.00
Ni (mg/kg)	13.00	0.14
Co (mg/kg)	<0.03	-
Cu (mg/kg)	<5.00	9.00
Mn (mg/kg)	<5.00	17.00
S (% TS)	0.22	0.10
Coliform bacteria (MPN/g)	4.6 × 10^3^	-
Mortality rate of ascaris eggs (%)	100	-

Mean standard error. *n* = 3. TS: Total solid; VS: Volatile solid; TN: Total nitrogen; TOC: Total organic carbon. C/N: Carbon nitrogen ratio.

**Table 2 ijerph-19-07256-t002:** The average contents of cellulose, hemicellulose, lignin, and total lignocellulose during the fermentation.

Erythromycin Concentrations (mg/L)	Lignin (% TS)	Hemicellulose (% TS)	Cellulose (% TS)	Total Lignocellulose (% TS)
0	16.07 ± 1.00	13.45 ± 0.53	16.14 ± 0.93	45.66 ± 1.58
1	16.88 ± 1.25	13.99 ± 0.42	15.51 ± 0.64	46.38 ± 1.16
20	19.66 ± 1.41 *	13.71 ± 0.62	14.91 ± 0.57	48.30 ± 1.18
40	21.07 ± 1.25 *	13.63 ± 0.68	14.86 ± 0.38	49.56 ± 1.10 *

Mean ± standard error. *n* = 9. * *p* < 0.05.

## Data Availability

Not applicable.

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
