# Peer review of "The Removal of Erythromycin and Its Effects on Anaerobic Fermentation"

_ijerph, 2022, doi:10.3390/ijerph19127256_

Round 1
Reviewer 1 Report
Dear Authors
Thank you very much for your manuscript submission. Your study is well-designed.
However, the presentation of your manuscript is too confusing and overwhelming.
- To avoid this problem, please do add a flow chart and schematic figures to your manuscript. These items help the reader to find out the procedures done in your work at one glance. In this regard, please do read and add the following paper to the References section of the manuscript. You can employ BioRender (https://biorender.com) for figures, ACD/ChemSketch (https://www.acdlabs.com/products/draw_nom/draw/chemsketch/) for drawing chemical structures and CmapTools (https://cmap.ihmc.us) for free flow charts as the materials of the following papers:
1.Writing a strong scientific paper in medicine and the biomedical sciences: a checklist and recommendations for early career researchers. Biol Futur. 2021 Dec;72(4):395-407. doi: 10.1007/s42977-021-00095-z. Epub 2021 Jul 28. PMID: 34554491.
- Erythromycin stimulates rather than inhibits methane production in anaerobic digestion of antibiotic fermentation dregs. Sci Total Environ. 2022 Feb 10;807(Pt 3):151007. doi: 10.1016/j.scitotenv.2021.151007. Epub 2021 Oct 16. PMID: 34666088.
- Pharmaceuticals effect and removal, at environmentally relevant concentrations, from sewage sludge during anaerobic digestion. Bioresour Technol. 2021 Jan;319:124102. doi: 10.1016/j.biortech.2020.124102. Epub 2020 Sep 16. PMID: 32977100.
- Combined effect of erythromycin, tetracycline and sulfamethoxazole on performance of anaerobic sequencing batch reactors. Bioresource technology. 2015 Jun 1;186:207-14.
2. Please add the used protocols in the Materials and Methods section.
3. There are some typos and grammatical errors. Please revise them!
4. Please do add Discussion section. The Results should be interpreted in Discussion section.
Author Response
Dear reviewer,
We really appreciate your useful comments and suggestions on our manuscript. According to the comments and suggestions, we have checked and modified the manuscript carefully. The detailed corrections are listed below. We have marked the changes in the revised manuscript using "Track Changes" function in Microsoft Word. We highly appreciate your attention and consideration on our work.
Sincerely,
Dr. Huayong Zhang
Point 1: Thank you very much for your manuscript submission. Your study is well-designed. However, the presentation of your manuscript is too confusing and overwhelming. To avoid this problem, please do add a flow chart and schematic figures to your manuscript. These items help the reader to find out the procedures done in your work at one glance. In this regard, please do read and add the following paper to the References section of the manuscript. You can employ BioRender (http://biorender.com) for figures. ACD/ChemSketch (https://www.acdlabs.com/products/draw_nom/draw/chemsketch/) for drawing chemical structures and CmapTools (http://cmap.ihmc.us) for free flow charts as the materials of the following papers:
- Writing a strong scientific paper in medicine and the biomedical sciences a checklist and recommendations for early career researchers. Biol Futur. 2021 Dec:72(4):395-407. doi: 10.1007/S42977-021 -00095-z. Epub 2021 Jul 28. PMID: 34554491.
- Erythromycin stimulates rather than inhibits methane production in anaerobic digestion of antibiotic fermentation dregs. Sci Total Environ. 2022 Feb 10;807(Pt 3):151007. doi: 10.1016/j.scitotenv.2021.151007. Epub 2021 Oct 16. PMID: 34666088.
- Pharmaceuticals effect and removal, at environmentally relevant concentrations, from sewage sludge during anaerobic digestion. Bioresour Technol. 2021 Jan;319:124102. doi: 10.1016/j.biortech.2020.124102. Epub 2020 Sep 16. PMID: 32977100.
- Combined effect of erythromycin, tetracycline and sulfamethoxazole on performance of anaerobic sequencing batch reactors. Bioresource technology. 2015 Jun 1;186:207-14.
Response 1: Thank you very much. A graphical abstract and a schematic figure are added according to the suggestions and comments. We have read the four papers you provided. We drew the flow chart and schematic figures according to the first paper. The remaining three papers have been added to the References section.
Point 2: Please add the used protocols in the Materials and Methods section.
Response 2: Thank you for the suggestion. We add an experimental flow chart and references for measurements in the Materials and Methods section.
Point 3: There are some typos and grammatical errors. Please revise them!
Response 3: Thank you for pointing out our carelessness. We checked the manuscript carefully and revised the errors.
Point 4: Please do add Discussion section. The Results should be interpreted in Discussion section.
Response 4: Thank you for the suggestion. We add discussion in section 3 Results and Discussion.

Reviewer 2 Report
- Objective should be more clearly written
- In Section 3, concentrations were studied like 0, 20 mg/L why so wide gap? Add another concentration at 10mg/L
- Recommendation should be provided for mechanism to stop the drug pollution
Author Response
Dear reviewer,
We really appreciate your useful comments and suggestions on our manuscript. According to the comments and suggestions, we have checked and modified the manuscript carefully. The detailed corrections are listed below. We have marked the changes in the revised manuscript using "Track Changes" function in Microsoft Word. We highly appreciate your attention and consideration on our work.
Sincerely,
Dr. Huayong Zhang
Point 1: Objective should be more clearly written.
Response 1: Thank you very much. Objective is clarified in Section 1 Introduction in the revised manuscript.
Point 2: In Section 3, concentrations were studied like 0, 20 mg/L why so wide gap? Add another concentration at 10mg/L.
Response 2: Thank you for your comments. We agree that if there was more concentration, the result will be more precise. However, the erythromycin concentrations at 10 mg/L may not change the results of the present study. On one hand, as shown in the results, erythromycin inhibited the fermentation process when the concentration is over 1 mg/L. There may be no threshold between 1 and 20 mg/L. 10 mg/L was seemed to inhibit the process too and the inhibitory effect was likely to be between 1 and 20 mg/L. On the other hand, the study was limited by the number of fermentation tanks. We would conduct more concentrations and more detailed studies in the future.
Point 3: Recommendation should be provided for mechanism to stop the drug pollution.
Response 3: Thank you for your suggestion. Recommendation has been added in Section 3.8 Implications of this study.

Reviewer 3 Report
The article describes an interesting work supported by enough experimental data. It could only be improved by including information on continuous fermentation tests . Otherwise, it is a well-written manuscript and the discussion is justified enough.
Author Response
Dear reviewer,
Thank you for your affirmation of our study. We agree that there may be new findings in the continuous fermentation experiment. We are planning to carry out continuous fermentation in the future.
Sincerely,
Dr. Huayong Zhang

Reviewer 4 Report
Title: The removal of erythromycin and its effects on anaerobic fermentation
1) Zhang et al., worked good in this paper. It is of great significance to study the mechanism of antibiotic removal based on anaerobic fermentation
2) Line 128: “The data in the study were the average of triplicate treatments “ … but no error bar was shown the on the data point on any of the figure. Please put error bars on each figure.
3) Please, rewrite conclusion section: We normally, do not write result values in conclusion rather statements based on the interpretation of results. In fact, you should restate the thesis and show how it has been developed through the body of the paper, recommendations if any and who can benefit from your research
Author Response
Dear reviewer,
We really appreciate your useful comments and suggestions on our manuscript. According to the comments and suggestions, we have checked and modified the manuscript carefully. The detailed corrections are listed below. We have marked the changes in the revised manuscript using "Track Changes" function in Microsoft Word. We highly appreciate your attention and consideration on our work.
Sincerely,
Dr. Huayong Zhang
Point 1: Zhang et al., worked good in this paper. It is of great significance to study the mechanism of antibiotic removal based on anaerobic fermentation.
Response 1: Thank you very much for your affirmation of our study.
Point 2: Line 128: "The data in the study were the average of triplicate treatments" ... but no error bar was shown the on the data point on any of the figure. Please put error bars on each figure.
Response 2: Thank you for your comments. We agree with your opinions that error bars should be shown when we clarify triplicate treatments. However, when we add error bars into the figures, the lines for different groups sometimes became unclear or even messy, for examples daily biogas yields (shown in the attachment), cellulase activities and so on. We wonder that could we keep the original figures to make sure the results clearer.
Point 3: Please, rewrite conclusion section: We normally, do not write result values in conclusion rather statements based on the interpretation of results. In fact, you should restate the thesis and show how it has been developed through the body of the paper, recommendations if any and who can benefit from your research.
Response 3: Thank you for your suggestion. The Conclusion section has been modified according to the suggestions in the revised manuscript.

Reviewer 5 Report
In this work the authors show a study on the elimination of the antibiotic erythromycin and its effects on fermentation.
Despite the large amount of methodology used by the authors and all the results they have obtained, the purpose of this work and the impact of their results are not very well understood. The results obtained in this work need to be better ordered and justified, indicating the relationship between them.
The methodology used and the results obtained seem to me to be adequate. However, I would ask the authors to rewrite the manuscript taking care of the objectives of the work presented. In addition, the results, discussion and conclusions obtained should be rewritten, in a clearer and more concise manner, to justify the execution and efficacy of the different studies performed.
I would also like to send a few minor suggestions:
- Lines 44, 46 and 104 should be written well CH4
- Table 1, line 70 4.6 x103 instead of 4.60x103
Author Response
Dear reviewer,
We really appreciate your useful comments and suggestions on our manuscript. According to the comments and suggestions, we have checked and modified the manuscript carefully. The detailed corrections are listed below. We have marked the changes in the revised manuscript using "Track Changes" function in Microsoft Word. We highly appreciate your attention and consideration on our work.
Sincerely,
Dr. Huayong Zhang
Point 1: In this work the authors show a study on the elimination of the antibiotic erythromycin and its effects on fermentation.
Despite the large amount of methodology used by the authors and all the results they have obtained, the purpose of this work and the impact of their results are not very well understood. The results obtained in this work need to be better ordered and justified, indicating the relationship between them.
The methodology used and the results obtained seem to me to be adequate. However, I would ask the authors to rewrite the manuscript taking care of the objectives of the work presented. In addition, the results, discussion and conclusions obtained should be rewritten, in a clearer and more concise manner, to justify the execution and efficacy of the different studies performed.
Response 1: Thank you very much for the suggestions and comments. We clarify the purpose of this work and the implications of the results in the revised manuscript. The results, discussion and conclusions are revised too.
Point 2: Lines 44. 46 and 104 should be written well CH4.
Response 2: Thank you for your comments. We checked the manuscript carefully and modified the typos.
Point 3: Table 1, line 70 4.6 ×103 instead of 4.60x103.
Response 3: We are sorry for our carelessness. We checked the manuscript carefully and modified the errors.

Round 2
Reviewer 1 Report
Dear Authors
Thank you very much for your effective revision. My decision regarding your work is "Accept".
However, please do revise the following cases:
1. On page 3, Line 104: flow chart is correct not flow chat
2. The CmapTools (http://cmap.ihmc.us) as an effective website for free flow charts is a material of the following paper. So, it is recommended to add the following paper to the References section of the manuscript, too.
Writing a strong scientific paper in medicine and the biomedical sciences a checklist and recommendations for early career researchers. Biol Futur. 2021 Dec:72(4):395-407. doi: 10.1007/S42977-021 -00095-z. Epub 2021 Jul 28. PMID: 34554491.
Author Response
Dear reviewer,
We really appreciate your useful comments and suggestions on our manuscript. According to the comments and suggestions, we have checked and modified the manuscript carefully. The detailed corrections are listed below. We have marked the changes in the revised manuscript using "Track Changes" function in Microsoft Word. We highly appreciate your attention and consideration on our work.
Sincerely,
Dr. Huayong Zhang
Point 1: Thank you very much for your effective revision. My decision regarding your work is "Accept". However, please do revise the following cases:
On page 3, Line 104: flow chart is correct not flow chat.
Response 1: Thank you very much. The typo has been modified.
Point 2: The CmapTools (http://cmap.ihmc.us) as an effective website for free flow charts is a material of the following paper. So, it is recommended to add the following paper to the References section of the manuscript, too.
Writing a strong scientific paper in medicine and the biomedical sciences a checklist and recommendations for early career researchers. Biol Futur. 2021 Dec:72(4):395-407. doi: 10.1007/S42977-021 -00095-z. Epub 2021 Jul 28. PMID: 34554491.
Response 2: Thank you for the suggestion. The abovementioned paper has been added to the References section.

Reviewer 5 Report
Dear authors
the revised version of the manuscript reads and understands much better.
Thank you very much for your modifications and revisions
This is a suggestion, the 0 should be eliminated (4.6 X 103) when writing 4.60 x 103
Author Response
Dear reviewer,
We really appreciate your useful comments and suggestions on our manuscript. According to the comments and suggestions, we have checked and modified the manuscript carefully. The detailed corrections are listed below. We have marked the changes in the revised manuscript using "Track Changes" function in Microsoft Word. We highly appreciate your attention and consideration on our work.
Sincerely,
Dr. Huayong Zhang
Point 1: The revised version of the manuscript reads and understands much better.
Thank you very much for your modifications and revisions
This is a suggestion, the 0 should be eliminated (4.6 × 103) when writing 4.60 × 103.
Response 1: We are sorry for our carelessness. The typo has been modified.
